# 4D Printing Self-Sensing and Load-Carrying Smart Components

**DOI:** 10.3390/ma17235903

**Published:** 2024-12-02

**Authors:** Yi Qin, Jianxin Qiao, Shuai Chi, Huichun Tian, Zexu Zhang, He Liu

**Affiliations:** 1School of Mechatronics Engineering, Harbin Institute of Technology, Harbin 150001, China; qy01111@sina.com (Y.Q.); 23s136388@stu.hit.edu.cn (S.C.); tianhuichun1220@163.com (H.T.); 23s136363@stu.hit.edu.cn (Z.Z.); 2Beijing Institute of Astronautical System Engineering, Beijing 100076, China; 3Research Institute of Petroleum Exploration & Development, Beijing 100083, China; 15163459196@163.com

**Keywords:** 4D printing, self-sensing, fused deposition modeling, intelligent component

## Abstract

In the past decade, 4D printing has received attention in the aerospace, automotive, robotics, and biomedical fields due to its lightweight structure and high productivity. Combining stimulus-responsive materials with 3D printing technology, which enables controllable changes in shape and mechanical properties, is a new technology for building smart bearing structures. A multilayer smart truss structural component with self-sensing function is designed, and an internal stress calibration strategy is established to better adapt to asymmetric loads. A material system consisting of continuous carbon fibers and polylactic acid was constructed, and an isosceles trapezoidal structure was chosen as the basic configuration of the smart component. The self-inductive properties are described by analyzing the relationship between the pressure applied to the specimen and the change in the specimen’s own resistance. Load-carrying capacity is realized by electrically heating the continuous carbon fibers in the component. Thermal deformation calibrates internal stress and enhances the load-carrying ability of the component over 50%. The experimental results demonstrate that the truss structure designed in this paper has strong self-induction, self-driving ability, and asymmetric load adaptation ability at the same time. This verifies that the 4D-printed smart component can be used as a load-carrying element, which broadens the application scope of smart components.

## 1. Introduction

Sensing external stimuli and actively transforming to adapt to changing environments is common in nature. Materials and structures with an active deformation function have aroused great enthusiasm among researchers. Among them, shape memory polymers (SMPs) show large deformability, excellent flexibility, and high processability, which have been widely employed in biological devices [1,2], medical materials [3], aerospace engineering [4], and other fields. With the rapid development of organic materials, various SMPs have been produced, such as polyurethanes, polyimides, polylactic acid, polycaprolactone, styrene–butadiene copolymers, epoxy resins, etc. The shape memory effect could be activated by heat field, electric field, light irradiation, magnetic field, chemical solvents, or other external stimuli [5,6].

The formation of intelligent components using an SMP is one of the important ways to optimize the existing bearing structure. Shape memory polymers usually have two forms, namely a temporary shape and an original shape. Under certain external stimuli, such as heat, electricity, light, magnetism or chemical solvents, etc., it can recover from the temporary shape to the original shape. Take thermal stimulation as an example. When the temperature is higher than its transition temperature (glass transition temperature Tg or melting point Tm), the polymer segments can move and deform under the action of an external force. When it is cooled below the transition temperature, the deformed shape is fixed, forming a temporary shape. When heated again above the transition temperature, the polymer segments can move again thus returning to the original shape. Continuous fiber-enhanced SMPs can further improve the poor mechanical properties of pure SMPs [7], the contributing factors of which have been widely studied [8]. Its low cost, multi-material composite, high precision, and low time cost make it preferable to prepare SMP composites. The combination of additive manufacturing and SMPs, also called 4D printing technology [9], has the ability to produce shape changes over time. It can be realized by stereolithography, fused deposition modeling (FDM), or digital light processing. The stress deformation exhibited by 4D-printed products include the following two parts: sensing external stimuli [10,11] and self-driving for deformation [12,13]. The combination of these two parts is undoubtedly valuable. He et al. [14] built a robot that can mimic the human grasping reflex, where the actuator responds differently according to whether it exceeds a threshold. Chen et al. [15] used 4D printing technology to realize dual-sensing function integrated materials, which can sense temperature and mechanical signals at the same time and can distinguish between the two signals under certain conditions. From a single sensor to the integration of multiple sensors, the self-sensing function of intelligent components has become more comprehensive. SMPs can realize self-actuation under temperature [16], voltage [17], light [18], magnetic [19], water [13], and other conditions.

The use of 4D-printed structures in areas such as space-deployable structures has received attention, but their use is currently limited to small, non-load-bearing parts because the available 4D-printed structures are not strong enough. If continuous fiber-reinforced composites are used for 4D printing, there is a significant improvement in the mechanical properties of the prints. Carbon fiber, with its excellent mechanical properties and high electrical conductivity, is a commonly used reinforcement in continuous fiber-reinforced 4D printing. The conductivity of carbon fiber is used to energize the prints when the prints themselves generate heat, i.e., Joule heat, and thus warm up. This controlled warming property can be used to direct the thermal deformation of 4D prints to fulfill some specific functions, but it is difficult to use these properties in load-bearing environments because it is difficult to control the deformation caused by the environmental conditions and the applied load. Therefore, this paper attempts to propose a method that deforms the print under a load but, at the same time, deforms it into a shape that is more suitable for carrying the current load, which is referred to in this paper as “adaptive load-bearing”. Adaptive load-bearing structures have a certain degree of flexibility and self-adjusting ability. When subjected to loads of different directions and magnitudes, they can redistribute stress through changes in their own material properties and internal structures, enabling the whole structure to better adapt to load changes and thus reducing the risk of damage caused by local stress concentration.

At the end of the last century, Chung et al. [20], who majored in composites, conducted the first electrodynamic tests on carbon fiber laminates under tension and found that a change in load led to a change in the resistance of the laminates, which indicated that the resistance of carbon fiber laminates was sensitive to the load. They also conducted many studies on the force-resistance effect of epoxy resin-based continuous carbon fiber bundle composites since 2000, and the results show that the resistance change in the composites has a good correspondence with the strain, and it is also found to be very repeatable after many cycles. Importantly, the sensitivity of this composite is much higher than the sensitivity of conventional strain gauges. Therefore, this project planned to fabricate carbon fiber self-sensing shells using a coaxial extrusion additive manufacturing method.

## 2. Methods

### 2.1. Model Establishment

The truss structure consisted of two parts, a beam of pure polylactic acid (PLA) resin material and a geometrically shaped support structure of continuous carbon fiber-reinforced PLA resin material, as shown in Figure 1. Polylactic acid was purchased from Zhuhai Sanlv Industrial Co., Ltd. (Zhuhai, China). Carbon fiber was purchased from Shenzhen Dongli Electronic Co. (Shenzhen, China). The temperature of the printing heat bed was 60 °C, the temperature of the print head was 200 °C, the printing layer height was 0.5 mm, and the moving speed of the print head was 7 mm/s. The part was printed with 100% infill. The dimensions of the printed part are shown in Figure 2.

In this paper, an adaptive support structure with sensing characteristics is proposed. The adaptive characteristics of models with various support shapes are evaluated. Samples are divided into three groups with six in each group for the experiment. The evaluation results are shown in Figure 3. The evaluation is based on the load size ratio before and after adaptation, denoted as follows:(1)$$\begin{array}{l} estimation=\frac{F_{after}}{F_{before}} \end{array}$$
where *F_before_* represents the maximum load-bearing pressure value of the component before adaptive treatment and *F_after_* represents the maximum load-bearing value of the component after adaptive treatment.

#### 2.1.1. Self-Sensing Mechanism

The piezoresistive effect, resistivity of the material changing with external force, can convert a dynamic signal to an electric signal, which has been widely employed to monitor pressure, stress, strain, velocity, and acceleration. Self-sensing of the intelligent component is also based on this mechanism. The piezoresistive effect can be expressed by the series expansion as follows:(2)$$p_{ij}=p_{ij}^{0}+\pi_{ijkl}\sigma_{kl}+\Lambda_{ijklmn}\sigma_{kl}\sigma_{mn}$$
where $p_{ij}^{0}$ is the resistivity tensor of material without external stress, *π_ijkl_* and *Λ_ijklmn_* are the fourth and sixth order tensors of resistivity induced by stress, and *σ_kl_* and *σ_mn_* are stress tensors, respectively.

When stress is at a low level, it can be regarded as a linear relationship between resistivity and stress, as follows:(3)$$p_{ij}=p_{ij}^{0}+\pi_{ijkl}\sigma_{kl}$$

Hence, resistivity tensor induced by external stress can be expressed as follows:(4)$$\Delta p_{ij}=\pi_{ijkl}\sigma_{kl}$$

The formula above is the piezoresistive constitutive equation of the material and can also be displayed in the following:(5)$$\Delta p_{ij}=C_{ijkl}\varepsilon_{kl}$$
where $C_{ijkl}$ is the fourth order tensor of resistivity induced by strain.

The piezoresistive coefficient exhibits the following characteristics: (1) Shear stress does not produce a positive piezoresistive effect; (2) positive stress does not produce shear piezoresistive effect; and (3) shear stress only produces a piezoresistive effect in its own shear stress plane. Therefore, the piezoresistive coefficient matrix can be expressed as follows:(6)$$\left[\begin{array}{cccccc} \pi_{11} & \pi_{12} & \pi_{13} & 0 & 0 & 0\\ \pi_{21} & \pi_{22} & \pi_{23} & 0 & 0 & 0\\ \pi_{31} & \pi_{32} & \pi_{33} & 0 & 0 & 0\\ 0 & 0 & 0 & \pi_{44} & 0 & 0\\ 0 & 0 & 0 & 0 & \pi_{55} & 0\\ 0 & 0 & 0 & 0 & 0 & \pi_{66} \end{array}\right]$$
where 1, 2, 3 are the axis directions of material and 4, 5, 6 are the shearing directions of the material.

According to the generalized Hooke’s theorem, as follows:(7)$$\left\{\begin{array}{c} \sigma_{x}=\frac{E}{\left(1+\mu \right)\left(1-2\mu \right)}\left[\left(1-\mu \right)\varepsilon_{x}+\mu \left(\varepsilon_{y}+\varepsilon_{z}\right)\right]\\ \sigma_{y}=\frac{E}{\left(1+\mu \right)\left(1-2\mu \right)}\left[\left(1-\mu \right)\varepsilon_{y}+\mu \left(\varepsilon_{z}+\varepsilon_{x}\right)\right]\\ \sigma_{z}=\frac{E}{\left(1+\mu \right)\left(1-2\mu \right)}\left[\left(1-\mu \right)\varepsilon_{z}+\mu \left(\varepsilon_{x}+\varepsilon_{y}\right)\right]\\ \gamma_{xy}=\tau_{xy}/G\\ \gamma_{yz}=\tau_{yz}/G\\ \gamma_{zx}=\tau_{zx}/G \end{array}\right.$$

The equation above can be expressed as a matrix as follows:(8)$$\left[\begin{array}{c} \sigma_{1}\\ \sigma_{2}\\ \sigma_{3}\\ \sigma_{4}\\ \sigma_{5}\\ \sigma_{6} \end{array}\right]=\frac{E}{\left(1+\mu )(1-2\mu \right)}\left[\begin{array}{cccccc} 1-\mu & \mu & \mu & 0 & 0 & 0\\ \mu & 1-\mu & \mu & 0 & 0 & 0\\ \mu & \mu & 1-\mu & 0 & 0 & 0\\ 0 & 0 & 0 & \frac{1-2\mu }{2} & 0 & 0\\ 0 & 0 & 0 & 0 & \frac{1-2\mu }{2} & 0\\ 0 & 0 & 0 & 0 & 0 & \frac{1-2\mu }{2} \end{array}\right]\left[\begin{array}{c} \varepsilon_{1}\\ \varepsilon_{2}\\ \varepsilon_{3}\\ \varepsilon_{4}\\ \varepsilon_{5}\\ \varepsilon_{6} \end{array}\right]$$

The tensor matrix of resistivity induced by strain can be obtained through the following:(9)$$\left[\begin{array}{cccccc} C_{11} & C_{12} & C_{13} & 0 & 0 & 0\\ C_{21} & C_{22} & C_{23} & 0 & 0 & 0\\ C_{31} & C_{32} & C_{33} & 0 & 0 & 0\\ 0 & 0 & 0 & C_{44} & 0 & 0\\ 0 & 0 & 0 & 0 & C_{55} & 0\\ 0 & 0 & 0 & 0 & 0 & C_{66} \end{array}\right]$$

In practical engineering applications, stress is difficult to measure effectively, so tensor matrix of resistivity induced by strain is emphatically analyzed in this work.

#### 2.1.2. Adaptive Bearing Mechanism

Polylactic acid (PLA) is a typical SMP, considering the mixture structure of the fixed and reversible phases. The crosslinked grids of the fixed phase act as a stabilizer for the permanent shape of the material, while the reversible phase is supposed to recombine the grid by a crosslinking reaction under certain temperature. The recombined temporary shape can be kept or changed by an external force, thus complex functions such as spreading and grasping become achievable. When the ambient temperature was reduced below the transition temperature, the reversible phase lost driving capacity and the component recovered to its initial configuration. Based on the mechanism above, several samples were built to probe the operating parameter ranges for self-actuating. In the application scenario proposed in this paper, the component was pressurized when the carbon fiber in the component was energized, because the carbon fiber itself has a certain resistance, so the component heated up and its own temperature rose, resulting in a certain deformation of the component structure under the action of the external load. This deformation under external load caused the component to show a “collapse” response to the stressed part, and this “collapse” created more contact area between the component and the object applying external load. At this time, we stopped powering the component to make it stop generating heat and allow it to cool down. The reversible phase of the cooled component lost its driving ability, and the component showed good rigidity again; the contact area between the component and the object applying the external load became larger. The stress on the component was reduced, and the load-carrying capacity was increased, which showed adaptive load-carrying characteristics. The specific process is shown in Figure 4.

### 2.2. Experimental Process

#### 2.2.1. Self-Sensing Experiment

The self-induction experimental process is relatively simple; its purpose is to explore the change in resistance of the test component as well as the influence of pressure on such component resistance. If the component resistance value and the amount of pressure have a certain linear relationship, the component can be used as the kind of support structure to sense the characteristics of that amount of pressure. The specific experimental process is as follows: The component will be printed and connected with a carbon fiber connector reserved for the resistance measurement circuit, along with a heat-shrinkable tube for protection. The assembled component is placed on the platform of the press and then slowly loaded to measure and record the changes in component resistance. The pressure test platform for the electronic universal testing machine can accommodate a loading speed of 1.5 mm/min, while the test machine has a loading head and component contact area of 200 mm^2^. The schematic diagram of the experimental device is shown in Figure 5.

#### 2.2.2. Adaptive Load-Bearing Experiments

In order to verify the adaptive loading characteristics of the components, the experiment uses the following printing process. Components of the same shape and material are divided into two groups—a group without adaptive loading treatment and a group with adaptive loading treatment—and the load-bearing capacity of the two groups are compared to verify the adaptive loading characteristics of the components with practical engineering value. The specific experimental process is as follows: Components are placed and fixed on the test bench before the start of the hydraulic press. The hydraulic press gradually applies pressure on the components until their destruction to record the size of the load at which such component destruction occurs. For the group of components that requires adaptive processing, the process is slightly more complex. The component is placed and fixed on the test bench. Then, the hydraulic press is started to apply a load to the component, during which the component is electrically energized such that it is heated and softened. The component deforms under loading and softening, at which point the hydraulic press stops loading and stops energizing the component to keep it deformed and allow it to cool. After the component is cooled and shaped, the hydraulic press starts up again and the application of the load is continued until the component is destroyed. The amount of load applied to the component at the time of destruction is recorded. The specific experimental steps are shown schematically in Figure 6.

## 3. Results

### 3.1. Experimental Results of Self-Sensing Characterization

At the end of the last century, D.D.L Chung, Xiaojun Wang, and their colleagues from Harburg University of Technology, Hamburg, Germany, who majored in composites, conducted the first electrodynamic tests on carbon fiber laminates under tension and found that a change in load led to a change in the resistance of the laminates, which indicated that the resistance of carbon fiber laminates was sensitive to the load. D.D.L Chung, Xiaojun Wang, and their team in the USA conducted many studies on the force-resistance effect of epoxy resin-based continuous carbon fiber bundle composites since 2000, and their results showed that the resistance change in the composites had a good correlation with the strain, and it was also found to be very repeatable after many cycles. Importantly, the sensitivity of these composites was much higher than the sensitivity of conventional strain gauges. Therefore, this project was planned to fabricate and study the carbon fiber self-sensing truss structure using a coaxial extrusion additive manufacturing method.

In the results of self-sensing experiments, the relationship between the change in sample resistance and the pressure applied to the sample is the main feature studied. The pressure–resistance curve clearly shows that the resistance of the sample increases with the increase in applied pressure. Scatterplots of the experimental results were plotted, and their patterns were fitted. A linear fit was applied to the relationship between the magnitude of the load applied to the sample and the resistance value; the goodness of fit was 0.9864, as shown in Figure 7.

The length of the geometrically supported portion of the assembly containing carbon fiber composites was 2320 mm, and the average resistance of the assembly in the static state was 990 Ω, with a resistivity of 8.3787 × 10^−5^ Ωm. The average value of the increase in resistance of the assembly during the process of compression to rupture was 207 Ω, with a rate of change of 20.9%. In order to verify the reliability of the self-inductive properties of the components in practical applications, a correlation analysis was performed to correlate the relationship between the resistance change in the components and the pressure applied to them. After fitting the experimental data, we found that there was a linear relationship between the pressure on the component and its resistance change, so Pearson’s correlation coefficient was chosen for the correlation analysis of these two variables, and the specific analysis process is as follows:

Two arrays—an array of resistance values, *R*, and an array of pressures on the component, *P*—were defined to record the size of the pressure on the component during the experiment and the size of the corresponding resistance values, respectively.
$$P_{P,R}=\frac{COV(P,R)}{\sigma (P)\sigma (R)}$$
where $P_{P,R}$ denotes Pearson’s correlation coefficient between the component resistance value and the magnitude of the pressure applied to the component, COV(P,R) denotes the covariance between the component resistance value and the magnitude of the pressure applied to the component, and σ(P) and σ(R) denote the standard deviation of the magnitude of the pressure applied to the component and the component resistance value, respectively. The result of the calculation is $P_{P,R}=0.993159355$, which proves that the correlation between the magnitude of the pressure applied to the component and the resistance value of the component is very good, and the self-sensing characteristics of the component proposed in this paper are reliable.

### 3.2. Adaptive Load-Bearing Experimental Results

The experimental results showed that the load-bearing capacity of the adaptively treated specimens was significantly increased, and the loads sustained by the adaptively treated components at the time of damage were significantly higher than those of the non-adaptively treated components. Subsequently, the angle of the component support structure was changed to perform the same comparison experiments, and the experimental results showed that the adaptive treatment of the components result in the characteristics of increased load-bearing capacity. The specific experimental results are organized as shown in Figure 8.

In the process of changing the angle of the component support structure to perform the same comparison experiment, we found that the 117° trapezoidal support component’s load-bearing capacity was enhanced by an average of 36%, the 120° trapezoidal support component’s load-bearing capacity was enhanced by an average of 59%, and the 135° trapezoidal support component’s load-bearing capacity was enhanced by an average of 64%; the bigger the angle of the trapezoidal support, the more significant the enhancement of the load-bearing capacity. At the same time, it is clear that the trapezoidal support angle of different components results in different load-bearing capacities, so the study of what kind of trapezoidal support angle allows for the optimal load-bearing capacity and enhancement of these components would be very valuable for engineering applications. The comprehensive support capacity of the component is expressed as the ultimate load that the component can withstand after adaptive adjustment, and the comprehensive support capacity of the 120° trapezoidal support component was found to be the most excellent among the three components tested in this paper.

In practical engineering applications, since the adaptive process of the component includes a softening process, the component temporarily loses its support ability. Thus, the components must be used in groups, such that when one component is affected by the pressure and undergoes adaptive deformation, the other part continues to bear the load, and when the component undergoing adaptive deformation regains rigidity, the other component can then undergo adaptive deformation, and so on, taking turns to complete the adaptive process. The specific process is shown in Figure 9.

## 4. Conclusions

The aim of this paper is to propose a 3D printable adaptive support structure with sensing properties. The structure can sense the pressure through the self-sensing function and undergo a certain shape change when approaching the point of destructive pressure in order to enhance its resistance to such destructive forces. The following conclusions can be drawn from this study:Continuous carbon fiber laminates still exhibit a significant force-resistance effect in 3D-printed truss structures, and this force-resistance effect can be used to achieve the perception of load size in 3D-printed structures.It is feasible to use the force of the load to drive the deformation of a support, such that the deformed support has a better load-bearing performance for the current load, and the adaptability of the support structure has a certain value.The strategy of using supporting components in groups, alternating load bearing, and implementing adaptive reinforcement in batches are feasible.

## Figures and Tables

**Figure 1 materials-17-05903-f001:**
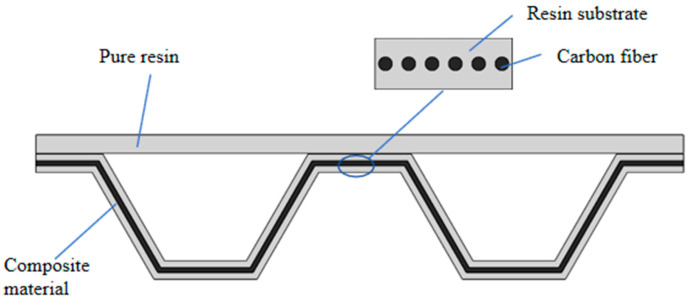
Schematic of the truss structure printed by continuous fiber-reinforced composite material.

**Figure 2 materials-17-05903-f002:**
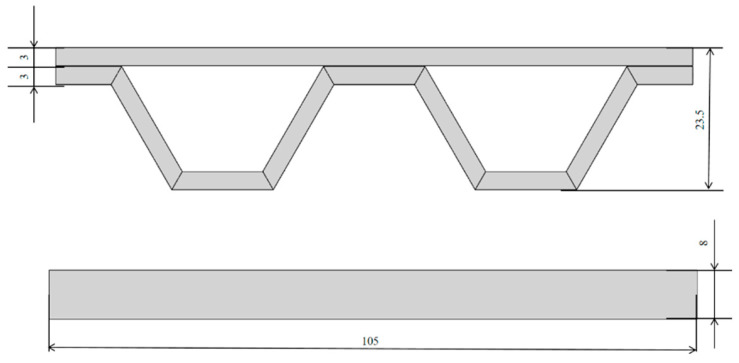
Schematic diagram of the size of the truss structure.

**Figure 3 materials-17-05903-f003:**
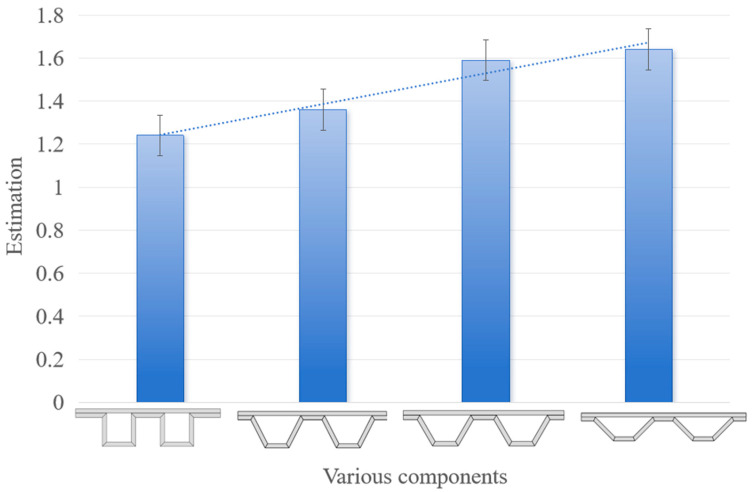
Comparison of adaptive load-carrying capacity of trapezoidal support structures from small to large bottom angles.

**Figure 4 materials-17-05903-f004:**
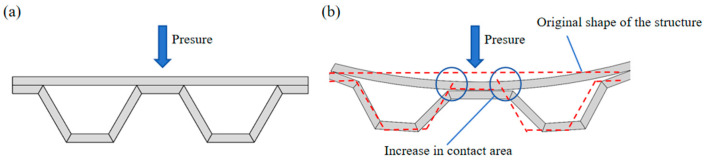
Schematic diagram of the adaptive deformation process. (**a**) External force; (**b**) Adaptive deforming structure.

**Figure 5 materials-17-05903-f005:**
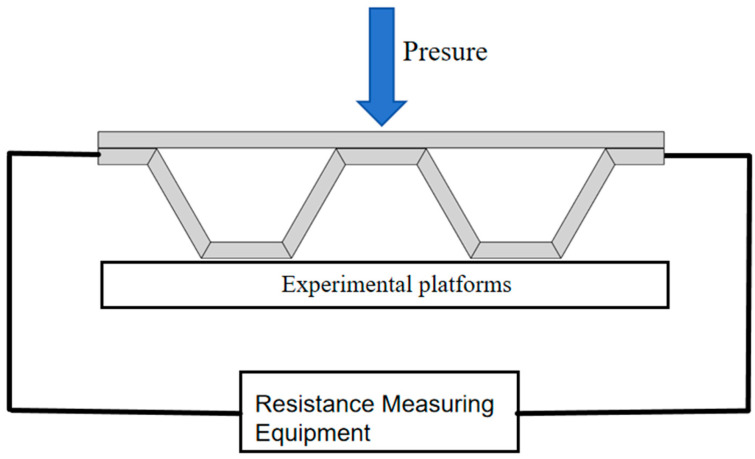
Schematic diagram of the component self-sensing experiment.

**Figure 6 materials-17-05903-f006:**
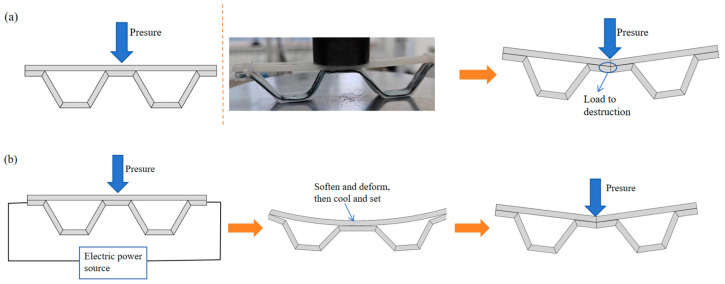
Schematic diagram of adaptive load bearing experiment. (**a**) Direct load-bearing; (**b**) Adaptive deformation load-bearing.

**Figure 7 materials-17-05903-f007:**
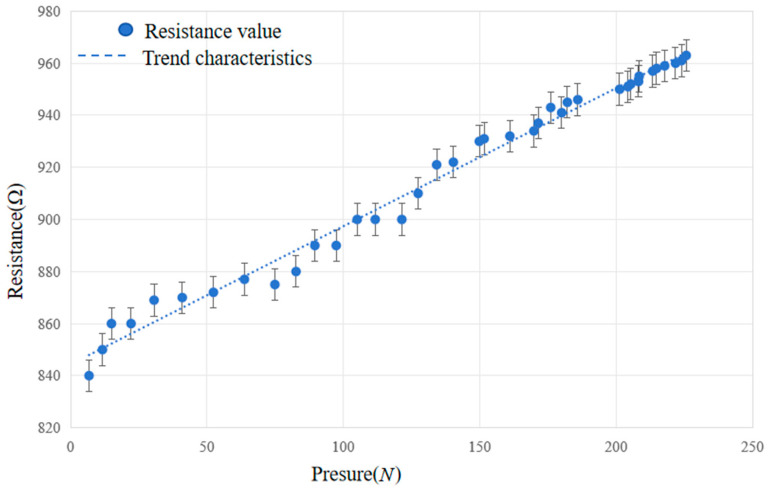
Variation in carbon fiber resistance with external pressure.

**Figure 8 materials-17-05903-f008:**
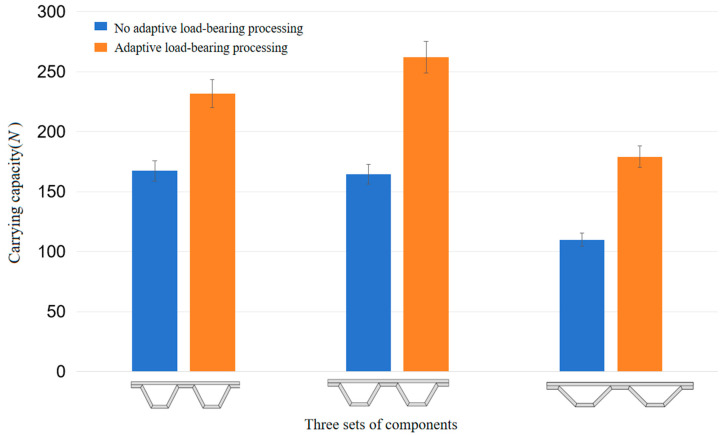
Load-bearing capacity of different component structures.

**Figure 9 materials-17-05903-f009:**
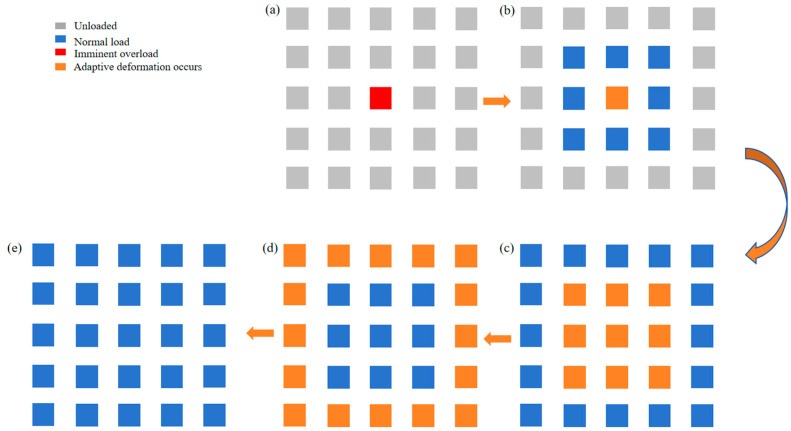
Schematic diagram of adaptive support strategy: (**a**) Partially loaded and about to be overloaded; (**b**) Peripheral components take over the load during softening of the loaded part; (**c**) The central area softens and the peripheral bearing; (**d**) Adaptive deformation of the central zone ends with successive softening of the periphery; (**e**) Overall adaptive deformation ends and carries.

## Data Availability

Data will be made available on request.

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
