# Peer review of "4D Printing Self-Sensing and Load-Carrying Smart Components"

_materials, 2024, doi:10.3390/ma17235903_

Round 1

Reviewer 1 Report

Comments and Suggestions for Authors

1. What are the dimensions of the part used in the tests? What are the relationships between the dimensions of the part and the results obtained?

2. What printing parameters were used for 3D printing of the part? For example, is the part hollow? The filling method directly influences the resistance and dynamics of the part under stress.

3. How many samples were used in the tests to conclude the reported correlation? Note that convincing data is needed to determine the signature of a part movement.

4. Is there any precedent for the chosen geometry? Why was this geometry chosen?

Author Response

We are very grateful to the reviewers for taking their precious time to give suggestions. We have revised the article according to the reviewers' comments. Please check the attachment.

Reviewer 2 Report

Comments and Suggestions for Authors

General Comment:

The paper introduces a 3D printable adaptive support structure with self-sensing capabilities, capable of detecting pressure and undergoing shape change to enhance its resistance when approaching destructive pressure. However, the work lacks a strong scientific approach, particularly in computational modelling and validation which would be valuable in predicting the structure's properties. Furthermore, the novelty of the work remains unclear and should be better articulated.

Specific Comments:

1.     Introduction section: the section would benefit from a clearer definition of key concepts like "adaptive load-bearing" at the beginning. While the introduction touches upon shape memory polymers (SMPs) and their applications, it lacks a clear distinction between SMPs and other smart materials. Including a brief comparison would provide better context for the reader.

2.     In the first paragraph of the introduction, the discussion of shape memory polymers focuses only on aerospace applications. It would be more comprehensive to include examples from other fields, such as biomedical and robotics, with appropriate citations like:

·       Rokaya, D., Skallevold, H. E., Srimaneepong, V., Marya, A., Shah, P. K., Khurshid, Z., ... & Sapkota, J. (2023). Shape memory polymeric materials for biomedical applications: an update. Journal of Composites Science7(1), 24.

·       McDonald-Wharry, J., Amirpour, M., Pickering, K. L., Battley, M., & Fu, Y. (2021). Moisture sensitivity and compressive performance of 3D-printed cellulose-biopolyester foam lattices. Additive Manufacturing40, 101918.

3.     The abbreviation "PLA" should be introduced the first time it is mentioned, rather than being defined later in the text.

4.     In Line 93, the terms "Fafter" and "Fbefore" should be written in subscript to improve readability and follow proper notation conventions.

5.     Figure 1: The dashed regression line in the graph should include the corresponding equation and R² value to provide a clearer understanding of the fit.

6.     Line 100: The definition of "piezoresistive effect" should be introduced earlier to avoid confusion.

7.     Methods Section: more clarity is needed on how the dimensions or geometric properties of the model impact its self-sensing and load-bearing capacity. Providing specific measurements and discussing their significance would add value. Including more detailed equations or computational simulations could strengthen the section.

8.     In Equation 1, the source or reference for this formulation should be cited to provide context.

9.     In Line 120, the reference to "He" should be clarified; it is unclear whether this refers to an author or something else.

10.  The inclusion of basic solid mechanics equations (Eq. 8-9) seems out of place here, especially without deeper contextualization. It would be more appropriate to focus on more advanced or relevant mechanics for this particular application.

11.  Figure 6: It would be more convincing if error bars or confidence intervals were included to show the variability in the measurements. This would give the reader a better understanding of the experimental reliability.

12.  There is a lack of comparison between the performance of the 4D-printed structure and conventional load-bearing components. Including quantitative comparisons of mechanical properties, or at least qualitative discussions of how this technology improves upon or complements existing solutions, would significantly enhance the paper's impact.

Comments on the Quality of English Language

need to improve

Author Response

(The authors gave the same response as above.)

Round 2

Reviewer 1 Report

Comments and Suggestions for Authors

Accept in present form.

Reviewer 2 Report

Comments and Suggestions for Authors

the authors successfully considered all my comments.